# Understanding ambivalence in help-seeking for suicidal people with comorbid depression and alcohol misuse

Milena Heinsch[1,2], Dara Sampson[1], Valerie Huens[3], Tonelle Handley[1,4], Tanya Hanstock[3], Keith Harris[5,6], Frances Kay-Lambkin [1,7] *

1 Priority Research Centre for Brain & Mental Health, University of Newcastle, Callaghan, NSW, Australia, 2 School of Social Work, Faculty of Arts and Humanities, University of Newcastle, Callaghan, NSW, Australia, 3 School of Psychology, Faulty of Science, University of Newcastle, Callaghan, NSW, Australia, 4 Centre for Rural and Remote Mental Health, Orange, NSW, Australia, 5 School of Psychology, Charles Sturt University, Port Macquarie, NSW, Australia, 6 School of Psychology, University of Queensland, Brisbane St Lucia, QLD, Australia, 7 National Drug and Alcohol Research Centre, UNSW, Randwick, NSW, Australia

* frances.kaylambkin@newcastle.edu.au

**Data Availability Statement:** Data cannot be shared publicly because of the potentially identifiable nature of the interview (qualitative) data, and ethics approval to provide access to the

## Abstract

Help-seeking prior to a suicide attempt is poorly understood. Participants were recruited from a previous research trial who reported a history of suicidal behaviours upon follow-up. Qualitative interviews were conducted with six adults to understand their lived experience of a suicide attempt and the issues affecting help-seeking prior to that attempt. Participants described being aware of personal and professional supports available; however, were ambivalent about accessing them for multiple reasons. This paper employs an ecological systems framework to better understand the complex and multi-layered interpersonal, societal and cultural challenges to help-seeking that people with suicidal ideation can experience.

## Introduction

Suicide is a prominent public health concern, which accounts for almost one million deaths per year worldwide and has devastating impacts on individuals, families and communities [1]. Suicide attempts constitute a major risk factor for completed suicide [2], however, research shows that help-seeking for suicidal ideation is low [3] and suicide prevention services are underutilised [4]. Although limited, prior research shows that negative attitudes and stigma relating to suicide and help-seeking behaviour result in lower intentions to seek help [3].

Suicide refers to the act of deliberately killing oneself [1]. A suicide attempt refers to the non-fatal attempt to inflict self-harm with the intent to die [5]. Suicidal ideation is a term used to refer to experiencing thoughts about suicide, which can be fleeting, can involve detailed planning, self-harm, and suicide attempts. Suicidal ideation can be assessed by determining frequency, intensity and duration of these suicidal thoughts [5]. Suicidal ideation is generally associated with depression; however, associations have been reported with many other

full interview transcripts was not obtained at the time of the study. Data are available from the University of Newcastle Human Research Ethics Committee (Contact: human-ethics@newcastle.edu.au) for researchers who meet the criteria for access to confidential data.

**Funding:** The authors received no specific funding for this work.

**Competing interests:** The authors have declared that no competing interests exist.

psychiatric disorders, life events, and family events, all of which may increase the risk of suicidal ideation [5].

Comorbid conditions, such as mental health and substance disorders, may present additional complexities and challenges to help-seeking by people experiencing suicidal ideation [6]. In particular, comorbid mood and substance disorders have been found to decrease help-seeking behaviours [6], suggesting that people with these comorbidities may require additional support when experiencing suicidal ideation. The urgency of this issue becomes even more apparent in light of the finding that depression and alcohol use disorders are the most common diagnoses in people with suicidal ideation, and that the risk of suicide increases exponentially when these disorders co-occur [7]; [8].

Suicide is often conceived of as a "funnel process" [9]. Individuals tend to first experience suicidal ideation and may then engage in planning ways to act on their ideation, leading to an attempt and in some cases suicide. However, contrary to the idea of suicidal thoughts and behaviours being on a continuum, research has shown that suicidal behaviours can be sporadic, and do not always occur in a progressive sequence. De Leo, Cerin, Spathonis and [10] used telephone interviews followed by a postal survey of 1,311 participants to determine the lifetime prevalence of suicidal ideation and attempts, and the possible development of suicidal behaviours on a continuum. They found that 57.1% ($n$ = 190) of participants identified that their suicidal thoughts fluctuated irregularly before they attempted suicide and that this was affected by co-morbid depression and alcohol use. These results indicate that, at least for this population, suicidal thoughts did not occur on a continuum of exacerbation, but that there are individuals who will continue to use suicide behaviour as a method for managing stressful life events. Further research examining the suicidal process, reckless behaviour, and help-seeking attitudes are therefore valuable to assist in the development of long-term prevention strategies and programs, as preventive interventions are possible at a number of points prior to suicide completion [9].

The interplay of broader social and environmental factors that impact on people's beliefs, attitudes and behaviours regarding help-seeking for suicidal ideation is complex. While the importance of multilevel, multifactorial systemic approaches to reducing suicide risk is beginning to be recognised in Australia [11]; [12], systems approaches to understanding and addressing suicide are in their infancy [13]. Accordingly, most studies have explored specific components of the help-seeking process, but very few have acknowledged or applied ecological approaches to a comprehensive exploration of the environmental influences (e.g., family, friends, and broader social networks) that appear critical in helping or hindering an individual's decision to seek help [14].

Bronfenbrenner's (1979)[15] ecological systems theory suggests that individuals are best understood within the context of their environment [16]. This theory provides a useful lens for understanding the multiple socio-cultural and political systems that surround and influence an individual. It also recognises the potential for individuals to influence their environment [17]. According to Bronfenbrenner (1979) [15], interaction between an individual and their environment occurs at multiple, interconnected levels, including in the microsystem, mesosystem, exosystem, macrosystem, and chronosystem (see Table 1). He argued that people are embedded within these multilayered systems, and that their development, behaviours and experiences ultimately result from their complex interactions with and between these systems.

While the ecological approach positions human development as intimately connected with, and dependent on, the multilayered contexts that surround them, it is important to acknowledge individuals' own agency within these systems. Early literature on this theoretical approach tended to overemphasise systemic factors, and somewhat neglected the role of individual difference [16]. Consequently, the theory was revised by Bronfenbrenner, introducing

**Table 1. Bronfenbrenner's (1974) ecological systems.**

| System | Explanation |
|---|---|
| Microsystem | The immediate environmental context in which an individual participates, and the people within this context, with whom the individual has direct contact; e.g. the family unit, school, work or other immediate social groups. |
| Mesosystem | The connections or influences between different elements of the microsystem; e.g. the intersection between work and family relationships. |
| Exosystem | The indirect or external influences on an individual from systems not directly related to, or connected with, the microsystem; e.g. the media, educational systems, community structures and legislation. |
| Macrosystem | Broader social, cultural and political influences/ideologies such as social and economic status, cultural values, beliefs, customs and laws. These underpin individual philosophies and behaviours, and filter throughout other systems of an individual's environment. |
| Chronosystem | Changes in an individual, their multi-layered systems, and all members of their environment across time. |

the term 'bio-ecological'[18], which acknowledges the influence of individual–alongside systemic–factors on a person's development and experiences.

The current paper reports on the findings of a qualitative study, which explored the experiences of people with comorbid depression and alcohol use disorders who had previously attempted suicide, and in particular, the help they sought and received prior to and following the attempt. The central aim of the study was to identify opportunities to encourage help-seeking by this population prior to potential suicide attempts. Applying an ecological systems theory lens to the discussion of findings, this paper also aims to present a deeper, more interconnected understanding of the multiple and broader systemic factors that may influence help-seeking behaviours for suicidality in situations where people are already experiencing complex life challenges.

## Methods

Participants for the current study were recruited by recontacting original participants from the Self–Help for Alcohol and other drug and Depression study (SHADE—[19]) to establish their history of suicide behaviour and help-seeking behaviour. As part of their participation in the original SHADE study, participants ($N$ = 274) gave extended consent to be contacted for further research projects (the SHADE project, Hunter Area Health Service HREC: 03/12/10/3.17, University of Newcastle HREC H-750-0204, Mid-Western Area Health Service HREC 2004/04, Central Coast Area Health Service HREC 04/30). Those who provided this extended consent formed the eligible pool for the current study, which received ethical approval from University of Newcastle (HREC reference H-2011-0335).

### Participants

Participants for the current study were recruited by recontacting original participants (N = 274) from the SHADE study (SHADE–[19], which tested a computerised treatment program for depression and alcohol/other drug use. Eligible participants in the original SHADE study were adults over 18 years of age who reported elevated depression symptoms (a score greater than 17 on the Beck Depression Inventory II [20] and concurrent use of alcohol in excess of national guidelines for low-risk consumption in place at the time of the study (four standard [10g ethanol] drinks per day for men or two for women) and/or at least weekly use of cannabis for the month prior to baseline. Exclusion criteria were active psychosis, inability to comprehend English sufficiently to understand the study interventions, and history of

traumatic brain injury severe enough to impair capacity to consent and participate in the study interventions. As part of the original SHADE study, all 274 participants were recontacted via mail after 5 years and asked to participate in a follow-up assessment (five-years following the original SHADE baseline). Participants received an information sheet, consent form, and an invitation to participate in the follow-up assessment. They were also advised that the follow-up assessment would include specific questions about previous suicidal thoughts and behaviours they may have experienced and that, if disclosed, they would receive an invitation to participate in an additional sub-study (the present study) about previous suicide attempts and help-seeking around those attempts. For the current study, eligible participants were those who provided consent to participate in the 5-year follow-up assessment for SHADE study participants, and who, during this assessment, indicated at least one previous suicide attempt. Participation in this sub-study was offered until it was determined that no new themes were emerging from the interviews.

## Procedure

Following provision of informed consent to participate in the qualitative sub-study a 30-minute semi-structured telephone interview occurred via the telephone, either at the time consent was provided, or at a subsequent time that was suitable for the participant. Participants were reimbursed $20AUD for their time and contribution to the study. The interview commenced with an open question about the participant's specific experience of their suicide attempt, and then probed for specific details around help sought and received at the time of the suicide attempt, allowing participants to initiate and discuss those aspects of their suicide attempt and associated help-seeking that were most salient to them (as per Braun and Clarke, 2006[21]). All interviews were audio recorded and transcribed verbatim by the interviewer (VH) immediately following the interview.

Interviews were conducted until it was determined that no new information or themes were observed in the data [22]. In order to reach this point in the study, each interview was reviewed for emerging themes by VH and TEH independently following each interview and before the next interview was scheduled. Once this was completed, VH and TEH met to discuss identified themes. This assisted in identifying new themes and determining when no new themes emerged. As an added measure of reliability in interview analysis, FKL was involved in review and interview discussions following three interviews (1st, 3rd and 6th). As a result of this process, a sample size of six was the point at which no new themes were emerging from the interviews.

## Measures

Of relevance to the current study are the following assessment measures that were collected for the 5-year follow-up assessment:

a. Beck Depression Inventory Fast Screen [23]: The BDI-FS is a 7-item self-report questionnaire used to screen for the presence of depressive symptoms. It is an affective measure of depression, while excluding symptoms potentially related to medical complications. Authors reported that scores 0–3 indicate minimal depression; 4–6 indicate mild depression; 7–9 indicate moderate depression; and 10–21 indicate severe depression.

b. Opiate Treatment Index [24]: The OTI was used in this study to measure the quantity and frequency of Alcohol, Cannabis and Tobacco use. Each subtype is assessed in terms of both quantity and frequency of use in the prior month to assessment. An average use quotient is calculated for the prior month, such that a sore of 1 equates to once daily use per day for the month prior to assessment.

c. General Help-Seeking Questionnaire [25]: The GHSQ was developed to assess intentions to seek help from different sources and for different problems. It uses a matrix format that can be modified according to purpose and need, therefore help sources and problem-types can be modified to meet sample characteristics and study requirements.

d. Suicide Behaviours Questionnaire-Revised [26]: The SBQR is a four-item assessment tool which assesses four domains of suicidality: lifetime attempts and ideation, suicide ideation in the last 12 months, the disclosure of suicidal behaviour and the self-reported likely hood of suicide behaviour.

## Analysis

Braun and Clarke's (2006)[21] six-phase model of thematic analysis was used in this study due to its accessible, theoretically flexible approach and potential to yield a 'rich and detailed, yet complex account of data' (p. 5). Thematic analysis was considered particularly useful for application in this study due to its 'theoretical freedom' [21]; [27], which made it suitable for use within the ecological systems theory framework that formed the basis of this study.

Qualitative data arising from the interviews were analysed using a combination of manual methods and NVivo 12. Using the best features of manual and electronic methods of analysis has been found to yield the best results in qualitative research [28]. Audio files were retained so the researcher could return to the recordings for a nuanced verification or clarification of content or meaning.

Initially, a manual thematic analysis was conducted independently by the student researcher (VH) and her two supervisors (TEH, FKL), through a brief reading of the six transcripts to identify recurrent patterns or themes within the data. This method was consistent with Braun and Clarke's (2006)[21] recommendation that "it is ideal to read through the entire data set at least once before you begin your coding, as ideas and identification of possible patterns will be shaped during the read through" (p. 87). In this study, the significance of a theme was determined not by quantifiable measures but rather by whether it captured something important in relation to the overall research question.

Following initial coding, each transcript was re-read and coded according to the preliminary codes established. Annotations were also made about possible connections between themes and additional themes. This stage involved the cross checking of coding strategies and interpretation of data by the student researcher and her two supervisors independently. The final stage of the analysis was to identify the 'story' that each theme tells and how this linked with the overall 'story' about the data [21]. This final stage was carried out by MH and DS.

## Results

There were six participants in this study (3 females and 3 males), and their background characteristics, based on their assessment from Part One of the study, are presented in Table 2.

As indicated in Table 2, participants ranged in age from 30–62 years, and reported depression scores between 0.00 (no current depressive symptoms) through to 8–9 (moderate depressive symptoms) through to 10 (severe current depressive symptoms). Only one participant (P4) indicated use of cannabis in the past month prior to the 5-year assessment (8 use occasions per day for the prior month), and three participants (P2, P3 and P6) indicated daily alcohol consumption in the previous month of between 8–14 standard drinks per day. Tobacco use ranged from minimal (P1, P3, P4 and P5) through to daily use in the prior month, of between 5–25 cigarettes per day.

**Table 2. Participant characteristics.**

| ID | Gender | Age | BDI FS* | Alcohol** | Cannabis** | Tobacco** |
|----|--------|-----|---------|-----------|------------|-----------|
| P1 | Female | 36 | 10.00 | 0.60 | 0.00 | 0.00 |
| P2 | Male | 44 | 8.00 | 14.75 | 0.00 | 12.00 |
| P3 | Male | 62 | 0.00 | 8.20 | 0.00 | 0.00 |
| P4 | Female | 29 | 5.00 | 0.36 | 8.00 | 1.00 |
| P5 | Male | 60 | 0.00 | 0.42 | 0.00 | 0.00 |
| P6 | Female | 39 | 9.00 | 10.40 | 0.00 | 25.00 |

*BDI FS = Beck Depression Inventory–Fast Screen score

**As measured by the Opiate Treatment Index. Scores indicate average use of each drug type in previous month to assessment. A score of 0.14 equates to once weekly use for the prior month; 1 is one use occasion per day for the previous month, 2 is two use occasions per day for the previous month, and so on.

All participants reported at least one previous suicide attempt as part of their eligibility for the current study. Details relating to these attempts are displayed in Table 3.

As indicated in Table 3, participants reported between 1 and 3 (P5) previous attempts, and none had previously told another person (friend, family member or health professional) of the impending attempt. Two participants reported suicidal thoughts in the past 12 months, ranging from once only (P5) to 3–4 times (P1).

Table 4 displays the self-reported help-seeking intentions of the participants from a range of professional and non-professional sources in relation to thoughts about suicide.

As Table 4 indicates, likelihood of help-seeking for suicidal thoughts was generally higher from professional than non-professional sources, with only one participant (P3) indicating he would be extremely unlikely to seek help at all for suicidal thoughts. Support from family/friends for suicidal thoughts ranged between extremely unlikely (P3) through to likely (P2), with no participant indicating this would be extremely likely for them. One participant (P2) indicated he would be extremely likely to seek support from a Minister or Religious Leader. Three participants reported they were extremely likely to seek support from a helpline for their suicidal thoughts (P2, P5, P6), however two also indicated they would be extremely unlikely to seek support from said helpline (P3, P4).

## Thematic analysis: Overview of themes

The participants willingly and openly described their suicidal experiences and their ability to seek help at the time of their attempt. Overall it was evident that participants were largely aware of the help available to them both through informal social networks and formal services. What was also very clear was a reluctance to access these supports in times of need for varying reasons including; affordability, access to, and awareness of services; compounding life events; coping mechanisms; and challenges of service engagement. Within these, sub-themes were identified, which were significant in describing the meaning associated with each key theme.

**Table 3. Suicide history of participants in the qualitative study.**

| ID | How many previous attempts? | Years of previous attempts | Thoughts of suicide in past year? | Ever told someone you were going to suicide? |
|----|------------------------------|-----------------------------|------------------------------------|----------------------------------------------|
| P1 | 2 | 1999, 2000 | 3–4 times | No |
| P2 | 1 | 1980 | Never | No |
| P3 | 1 | 2000 | Never | No |
| P4 | 1 | 2008 | Never | No |
| P5 | 3 | 2000, 2003, 2014 | Once | No |
| P6 | 2 | 1979, 2006 | Never | No |

**Table 4. Likelihood of help-seeking for suicidal thoughts from a range of sources.**[*]

| ID | Family/Friend[**] | Mental Health Professional | Phone Helpline | GP | Religious Leader | Would not seek help |
|----|----|----|----|----|----|----|
| P1 | 2.50 | 7.00 | 3.00 | 7.00 | 1.00 | 3.00 |
| P2 | 5.50 | 4.00 | 7.00 | 4.00 | 7.00 | 1.00 |
| P3 | 1.00 | 1.00 | 1.00 | 1.00 | 1.00 | 7.00 |
| P4 | 3.25 | 7.00 | 1.00 | 7.00 | 1.00 | 4.00 |
| P5 | 4.00 | 7.00 | 7.00 | 7.00 | 1.00 | 1.00 |
| P6 | 2.50 | 4.00 | 7.00 | 1.00 | 1.00 | 3.00 |

[*]Participants rated each source of help on the same scale from 1 (extremely unlikely), 3 (unlikely), 5 (likely) to 7 (extremely likely).

[**]Scores averaged for each participant across: intimate partner, friend, parent, other relative.

**Reluctance to seek help.** A key theme identified in this study was participants' reluctance to seek help for suicidal thoughts and behaviours from both family and friends and from community-based services. There was an overarching sense that participants gained little benefit from seeking help within their immediate social network of family, friends, or close relatives. Two key reasons were identified for this. Firstly, participants indicated that their social supports were judgmental, unsupportive, and unresponsive to cries for help:

"I was just attention seeking as far as they were concerned. . . something happened and I ended up at the front of my place in a crumbled heap on the ground out of frustration, my two sisters and brother, were there and they just looked at me and laughed and drove off." (P1, female, age 36 years).

For some, this lack of support was connected to the stigma of mental health:

"Well it's the way people treated you. . . what would have helped? Not having [suicidal thoughts] and . . . not having the stigma that people put on having mental problem." (P1, female, age 36 years)

Participants also expressed concern that a request for help may place an undue burden on family and friends, who may not be emotionally equipped or prepared for such a role:

"The burden of 'I'm going to finish it' is a bit too much of a burden to lay on a friend or intimate partner. Especially an intimate partner, I mean I can share things of life with my partner but the deep seeded emotions. . . no. You can't be openly transparent even with the closest. . .person you are with" (P2, male, age 44 years).

Several participants who did seek help from a friend felt that the high level of distress this caused placed pressure on the relationship, which in turn, decreased the likelihood that they would share these thoughts with friends or family in the future.

"Most probably not [seek help from a friend again] because it freaked her totally out, and it's not a nice thing to put on someone, I don't think." (P4, female, age 29 years).

One participant (P5) spoke highly of the support he received from his friend after his suicide attempt. This participant expressed gratitude that his friend persisted with her attempts to engage him with professional assistance at the peak of his distress:

"He [Psychiatrist] had the order from the Court to say that I had to be released [from inpa-tient unit] into her [friend] care . . . she was there to sort of manage me, if you like . . . and she sort of. . .she's put me on the right track" (P5, male, age 60 years)

Participants' responses reflected the complexity of their decision making in relation to who they would seek help from. While some recognised the value of talking with a trained profes-sional who is less involved, they also indicated that they would find it easier to confide in some-one they feel close to:

"Maybe a trained professional is better. . . someone that's not so close. . .but I find it easier talking to a close friend." (P6, female, age 39 years)

In relation to engagement with community support services, no participants had engaged with a service as their first point of contact for help for suicidal thoughts, although some partic-ipants were linked into mental health services at the time. Of these participants, none discussed their suicidal thoughts directly with their treating professionals, with one participant choosing to communicate via his friend instead:

"I spoke to my friend about it, and not with any of the professionals or anything, and my friend went around and spoke to the psychiatrist." (P5, male, age 60 years).

One reason cited for the lack of communication with service providers was an impression that available providers did not offer adequate, proactive support, particularly once people had been discharged into the community:

"We looked after my dad, he had a stroke, and when he died apparently I had a nervous breakdown. Then they put me in a mental home and then I came up here [moved house]. The mental health [service] came and assessed me, said they'd be back and I never ever saw then again or heard from them or nothing." (P4, female, age 29 years).

One participant noted that she had intended to disclose her suicidal thoughts to her doctor but did not feel safe to do this in the presence of his medical students:

"I had to go for my appointment with [service provider] and he had students in the room and I didn't want to see him with students . . .didn't want to talk about it in front of them. . .I left and went home" (P1, female, age 36 years).

Importantly, there were cases where participants were linked into public community-based services post-attempt, and these participants were more likely to engage in help-seeking for future suicidal crises:

"Yeah. . . I really hooked into mental health. . .they have saved my life" (P3, male, age 62 years).

An interesting finding was that help-seeking was associated with a lack of intent to commit suicide, with one participant noting that if he really wanted to commit suicide, he would not tell anybody:

"If I was going to attempt suicide right now, I wouldn't tell anybody because that to me means that I don't really want to commit suicide." (P3, male, age 62 years).

**Affordability, access and awareness of services.** Affordability, access to, and awareness of, services were identified as barriers to help-seeking. For example, one participant described living alone in an unfamiliar location where help was difficult to access:

"I was living on my own in a strange town. . .[help was] an hour away" (P4, female, age 29 years).

Another participant highlighted the perceived cost of therapy as an inhibiting factor to help-seeking, combined with a sense that support was not available anyway:

"Couldn't afford it. And there wasn't any really here. You know like there was no help. . . and it's still the same." (P4, female, age 29 years).

Lack of awareness of available support was also related to age, with one participant noting that he was young and did not know about existing support options:

"Well maybe they were in place. . . but I was pretty young, and I was unaware of other help" (P2, male, age 44 years).

Availability of services did not always enhance help-seeking, with one participant reporting that living in a small rural town prevented the anonymity she needed to access available help:

"I mean. . . just if there was a psychiatrist in [town] . . . but then everybody would have known about. . .it's one of those little towns that everyone knows everyone" (P6, female, age 39 years).

**Compounding life events and coping mechanisms.** Participants all reported experiencing adverse and compounding life events immediately prior to their suicide attempt, which placed a high level of stress on them, leading them to contemplate suicide as an escape route:

"Yeah I had three young children and was bringing them up on my own. The situation had arose, I had no family to turn to, my youngest sister accused my ex-husband to have interfered with her as a child . . . I had only just really recovered from a bad marriage, and severe chronic back pain and surgery on my back. I was trying to cope with all of that and still work and look after my children. . . so it was like I don't want to be here right now without thinking of the consequence." (P1, female, age 36 years).

Participants' decision to attempt suicide was often influenced by a sense that there was no solution to, or way of escaping, their current situation. For example, one participant described his suicide as a response to his fear of the future as a young man:

"You fear leaving your job but you also fear going forward. . .the story goes. . .overdose of pills and I drove my car into the bush until I crashed and went to sleep in the back seat and two hours later the sun was blaring in my face and I went oh that didn't work. . . . I went home and told my parents this is what I did. . . I tried to commit suicide and they went yeah right you stupid young man. . . This is what I went through with my counsellor as well, in an odd way I still probably had pent up feelings about that until probably my 40's." (P2, male, aged 44 years).

Some participants reported using alcohol as a coping mechanism at times of significant stress, noting that this hindered them from addressing the difficulties they were experiencing:

"I didn't know what was going on with me. And I was using alcohol as an excuse to, you know, just drown everything out, which was bad." (P4, female, age 29 years).

**Need for proactive, flexible support.** While a number of participants demonstrated good knowledge of available services, many expressed ambivalence about engaging with services due to prior negative experiences with service providers, or fears about how a service provider might respond to a disclosure of suicidal ideation. However, participants often reported that, while they would not have sought help themselves, they would have accepted it had it been offered. They emphasised the importance of proactive, sensitive service engagement during a suicidal crisis. However, they also expressed a need to be self-reliant once the immediate crisis had passed.

*Initial support followed by self-reliance.* Participants who received initial support during a suicidal crisis often reported withdrawing from this support once the crisis had passed. These participants continued to make use of the materials and skills provided to them, suggesting that brief interventions and information provision might be important modes of support for people experiencing suicidal crisis:

"I ended up, I did, you know the Cognitive [Behaviour] Therapy. Well you have the paper-work, you can always read back over it, or you know stuff like that. . . . And that was, that was good. Well I was seeing like a counsellor, you know, where she sits and talks to you and gives you other opinions and ideas. You know how to go about things different ways. . . .. And then I was supposed to go back and see her because I was a bit all over the place, and I just couldn't have been bothered. My doctor said like why haven't you gone back? I said "Oh I couldn't have been bothered, was over it." . . . But she's there you know if I needed to go and see and talk to her. I try and just use what she taught me, work it out, sometimes it doesn't" (P4, female, age 29 years).

The importance of the initial response was highlighted by participants who described a single comment or insight that had helped them to move forward after a suicide attempt. For example, one participant reflected on a conversation with her son in which she was reminded of the impact a suicide would have on her family and grandchild:

"I was happy they [Emergency Department] didn't put me in the psych unit. . . . No [I didn't seek help]. I just woke up to myself when my son said I had the photo of my grand-daughter with me when I tried it [suicide] and I thought well, you know this is just bloody selfish, I can't do it." (P6, female, age 39 years).

Some participants did report receiving a more extended response from mental health professionals. However, this support was generally associated with higher levels of crisis, while initial help-seeking tended to be directed towards friends and regular service providers like the GP:

"Well I thought it was a bit funny at first because I'd walk in and sit down with the guy [psychologist] and he'd be, 'how'd you go, what have you been doing, catching any fish' or whatever. I didn't realise at the time he was just getting me relaxed and then he'd say 'you

haven't felt like this have you' . . .. I never realised that I'd given him the answer and then he'd say, 'well last time when I asked you about that you said so and so, this time you've said, so and so, that's an improvement on this' you know. Yeah, that's right he set me up really well. . . .Oh yeah, yeah, if I ever got down I'd go and see him but, you know, my first port of call, other than my two friends is my doctor. . .' cause he's really taken me under wing" (P5, male, age 60 years).

*Need for proactive service engagement.* Several participants highlighted that their most critical need for service engagement and support was during the initial crisis stage, and that they felt services did *"not try hard enough"* (P4) to engage or intervene with them at this time. Where support was not offered during a crisis, participants reported experiencing adverse consequences:

"The first time I was seeking help I had an appointment to see a counsellor, I think it was, I was very distraught, I had held myself together until I got to that appointment. I turned up and I was a day early. And I was so stressed and I said 'but I need to see someone now'. . . but they said no not until tomorrow. I went home and said how am I going to get through 'til tomorrow? I took a couple of valium to help me get to sleep and before I knew it I had taken the whole packet in that state." (P1, female, age 36 years).

Some participants perceived a lack of service response during a crisis as a clear indication that the service did not care, and this discouraged any further attempt to seek help:

"Because when I rang up like and said to them you know "you came out and assessed me. You said you'd be back. You haven't come back. You have given me nothing. You've just got me hanging, and I don't know what's wrong with me". And they [service providers] said, "Oh that's not our problem, we're booked out." . . .they didn't give a sXXX really . . .I won't do that again" (P4, female, age 29 years)

Several participants emphasised the importance of receiving an offer of help at times when they were not able to ask for support:

". . . there is no one I would have asked. . .it was easier to end it then tell people because you felt stigmatised . . .and they [services] didn't offer either. . .really other people should have seen the writing on the wall but didn't." (P3, male, age 62 years).

Participants often reported that while they would not have sought help themselves, they would have accepted it had it been offered:

"I just didn't think of it, I didn't want to go to the doctors. And really others should have seen the writing on the wall, but I didn't . . .If someone had offered a hand to me, I would have taken it." (P3, male, age 62 years)

## Discussion

This study aimed to explore the help-seeking experiences and attitudes of individuals with comorbid depression and alcohol use who had a suicide attempt in their lifetime. Findings show that the relationship between help-seeking behaviours and suicidal thoughts and attempts is a complex one, involving internal conflict between an individual's perception that

they "*should*" manage these thoughts and behaviours on their own without "*burdening*" family and close friends, while at the same time wanting family and friends to understand them, and to "*offer*" support. This complexity was also apparent in relation to service engagement, with many participants expressing a belief that professionals and services have a significant role to play in preventing a suicide attempt, while also emphasising a need for self-sufficiency and independence once the immediate crisis had passed. These findings suggest that individuals at risk of suicide require support that is both flexible and occurs at multiple systemic levels.

In particular, findings reveal the importance of communication between, and amongst, the people and services who inhabit the various systems surrounding an individual at risk of suicide–their mesosystem. For example, one participant reported a preference for communicating with treating professionals via his friend. Others noted that they would seek help from friends and family in the first instance, suggesting the microsystem has a crucial role to play in linking individuals experiencing suicidal ideation with support services. Participants who were encouraged to access services in this way, were more likely to engage in help-seeking for future suicidal crises. These findings reflect the need for a 'circle of support', in which family, friends and professionals work together to offset a suicide attempt. The notion of a circle of support confirms previous research findings, which identified the importance of social connectedness and support networks for reducing suicide risk [29].

The finding that individuals experiencing a suicidal crisis are more likely to seek support from family and friends in the first instance, is consistent with ecosystems theory, which emphasises the primacy of a person's microsystemic interactions [30]. According to Rogoff (2003)[31] the microsystem exerts a more powerful influence on individuals than any other contextual factors. Interactions that occur at this level have the power to be either extremely beneficial, or detrimental, to a person's development and wellbeing. This perspective usefully illuminates participants' ambivalence about seeking support from family, friends and support services, in an effort to avoid the damaging personal consequences of a judgemental, unsupportive response, at a time when they are arguably at their most vulnerable. Conversely, participants who encountered positive responses within their circle of support indicated that this had a crucial impact on their decision not to attempt suicide in the future.

Participants in this study expressed a need for proactive service engagement during a suicidal crisis. However, many individuals did not disclose their suicidal thoughts to health professionals they were in contact with prior to their suicide attempt. A central reason for this was a belief that services either did not recognise the seriousness of the situation or did not care enough to offer support. This suggests that it might be important for professionals to consider routinely asking suicide risk assessment questions of all clients who are engaged with health services (acute, community), irrespective of perceived risk, as this may have important preventative implications. Privacy was revealed as an important consideration in this context, with one participant noting she had planned to disclose to her doctor but decided not to do so in front of medical students. This highlights a need for service providers to ensure that their provision of professional development opportunities does not impact on service-user wellbeing, and to employ a high level of sensitivity when including trainees in routine service provision.

A further finding was that adverse and compounding life events significantly impacted participants' level of stress. For some, this led to an increase in unhealthy coping mechanisms, such as alcohol use, which presented a further barrier to addressing challenges or accessing support. This supports findings from previous research that comorbid conditions such as substance disorders present additional complexities and barriers to help-seeking for people experiencing suicidal ideation [6]. From an ecological systems perspective, the influence of an individual's microsystem extends beyond their interactions with other people, to their engagement with objects and symbols [32]. It is therefore possible to infer that, in the absence of

support from significant others, an individual may seek comfort in non-human objects such as drugs and alcohol.

This study revealed the ambivalence that people who are acutely suicidal can experience when engaging with different systems of support, with several participants expressing fears of negative, stigmatising reactions by family members or formal providers to a disclosure of suicidal ideation. This supports Calear et al.'s (2014)[3] finding that perceived negative attitudes and stigma relating to suicide and mental illness can lead to a reduction in help seeking intentions and behaviours. An ecological systems perspective usefully illuminates the critical role of broader social and cultural—macrosystemic—factors in shaping people's attitudes and behaviours to suicidality. The macro-system constitutes the shared belief systems and values of a cultural group [32]. This system has a cascading influence on interactions at all other systemic levels [33]. Employing this perspective, it is possible to observe the continuing legacy of the historical stigmatisation of suicide as a 'taboo' subject [34] and a 'sinful' act [35] at the individual level. In this way, an ecological systems frame may enhance our critical awareness of the wider socio-political context that shapes people's experiences of, and responses to, suicidal ideation [17]; [16].

## Limitations

The present study has several limitations that should be considered when evaluating the presented conclusions and implications. The authors acknowledge the difficulties in determining true data saturation [36] particularly when working with such small sample sizes [37] as in the current study. It is uncertain if other participants might have provided differing themes and experiences. It is a limitation that the exclusion criteria extended to people who could not speak or understand English sufficiently to engage in the study, highlighting further the potential issues with generalising these results to diverse groups in the community. Even though a randomised approach to the interview schedule was applied it is uncertain if there could be a bias in the sample pool. In addition, although rigorous standards for interpreting and analysing the qualitative data were applied, these are still subject to the researchers' experience and subjectivity may have affected the results.

## Conclusion

This study revealed the complex challenges people who are acutely suicidal experience when they engage with different systems of support. Findings highlight important considerations for friends and family members, who are often the first point of call for a person seeking support. They also emphasise the need for sensitive and proactive service engagement with individuals experiencing a suicidal crisis, to avoid the damaging impacts of inaction, stigma and judgement. The qualitative data revealed themes that illustrated many challenges of engaging with different support systems which lends itself well to an ecological systems theory perspective when considering interventions and approaches for this complex population.

## Acknowledgments

The authors would like to acknowledge the work of the researchers involved in the original study from which the participant sample was drawn. They would also like to thank the research assistance provided by Kellie Cathcart and Julia Rosenfeld. The researchers also thank the participants in the study for their time and openness to discussing this important, but sensitive, topic.

## Author Contributions

**Conceptualization:** Frances Kay-Lambkin.

**Data curation:** Valerie Huens, Keith Harris, Frances Kay-Lambkin.

**Formal analysis:** Milena Heinsch, Dara Sampson, Valerie Huens, Tonelle Handley, Frances Kay-Lambkin.

**Investigation:** Valerie Huens, Frances Kay-Lambkin.

**Methodology:** Milena Heinsch, Dara Sampson, Valerie Huens, Tonelle Handley, Keith Harris, Frances Kay-Lambkin.

**Project administration:** Frances Kay-Lambkin.

**Resources:** Frances Kay-Lambkin.

**Supervision:** Tanya Hanstock, Keith Harris, Frances Kay-Lambkin.

**Writing – original draft:** Valerie Huens, Tonelle Handley, Tanya Hanstock, Keith Harris, Frances Kay-Lambkin.

**Writing – review & editing:** Milena Heinsch, Dara Sampson, Valerie Huens, Tonelle Handley, Tanya Hanstock, Keith Harris, Frances Kay-Lambkin.

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
