## [Decision Letter · Decision Letter 0]

5 Aug 2019

PONE-D-19-16846

The paradox of engagement: The support needs of people with comorbid depression and alcohol misuse who had previously attempted suicide

PLOS ONE

Dear Professor Kay-Lambkin,

Thank you for submitting your manuscript to PLOS ONE. After careful consideration, we have decided that your manuscript does not meet our criteria for publication and must therefore be rejected.

I am sorry that we cannot be more positive on this occasion, but hope that you appreciate the reasons for this decision.

Yours sincerely,

Vincenzo De Luca

Academic Editor

PLOS ONE

Reviewers' comments:

Reviewer's Responses to Questions

**Comments to the Author**

1. Is the manuscript technically sound, and do the data support the conclusions?

Reviewer #1: No

2. Has the statistical analysis been performed appropriately and rigorously? 

Reviewer #1: No

3. Have the authors made all data underlying the findings in their manuscript fully available?

Reviewer #1: No

4. Is the manuscript presented in an intelligible fashion and written in standard English?

Reviewer #1: No

5. Review Comments to the Author

Reviewer #1: This manuscript provides an in-depth report of interviews conducted for six individuals who have previously attempted suicide to explore the relationship between depression, previous alcohol use and barriers to obtain help. The first author appears to be a student who has a great interest in suicide research and a compassion for those she has interviewed.

Although the enthusiasm for the topic is clearly present, the scientific rigor needed to contribute useful information for future work on prevention and treatment is totally absent. First, a sample that is 10-100 times larger would be needed in order to include a sufficient number of participants of varying gender, ethnicity, socioeconomic status, and access to health care providers, to name just a few of the variables that would appear to be important, to evaluate the hypotheses the authors wish to test. It is critical that any new research provide clear and convincing evidence that the sample studied is representative of the geographic and demographic areas of interest. This manuscript does not do this.

The instruments used are not adequate for the hypotheses to be tested. There are no standard measures of alcohol use, for example, such as the AUDIT. Without this standardization, the results obtained in this study cannot be put into the context of other studies. Use of the Beck Depression Inventory is standard for assessing depression as a current state. However, the study suffers from lack of information about what lifetime psychiatric diagnoses the participants have. One would expect the access to empathetic treatment providers would differ greatly among those with psychotic depression and those without as just one example.

6. PLOS authors have the option to publish the peer review history of their article (what does this mean?). If published, this will include your full peer review and any attached files.

Reviewer #1: No

- - - - -

---

## [Author Response · Author response to Decision Letter 0]

9 Sep 2019

Thank you for the opportunity to formally appeal the rejection of our manuscript (PONE-D-

19-16846). Please see below our point-by-point response to the reviewer:

Reviewer #1: Although the enthusiasm for the topic is clearly present, the scientific rigor

needed to contribute useful information for future work on prevention and treatment is totally

absent. First, a sample that is 10-100 times larger would be needed in order to include a

sufficient number of participants of varying gender, ethnicity, socioeconomic status, and

access to health care providers, to name just a few of the variables that would appear to be

important, to evaluate the hypotheses the authors wish to test. It is critical that any new

research provide clear and convincing evidence that the sample studied is representative of the

geographic and demographic areas of interest. This manuscript does not do this.

Response: Firstly, the reviewer’s comment regarding sample size (“a sample that is 10-100

times larger would be needed”) does not align with standards of rigour in qualitative research,

where sample size recommendations for studies using qualitative interviews range from five-

25 on average for HDR studies (Mason, 2010). Further, the reviewer’s assumption that we

were seeking to test a hypothesis, or to generalise our findings to the larger population, is

inconsistent with the stated aims of our paper (“to present a deeper, more interconnected

understanding of the multiple and broader systemic factors that may influence help-seeking

behaviours for suicidality”) or the exploratory nature of qualitative methodologies, which do

not intend to provide conclusive answers, but to gain a more in-depth understanding of the

complex nature of the issue under investigation. Including a more detailed statement about the

limitations of the sample size, and the subsequent lack of generalisability, in manuscript

would address this issue.

Reviewer #1: The instruments used are not adequate for the hypotheses to be tested. There are

no standard measures of alcohol use, for example, such as the AUDIT. Without this

standardization, the results obtained in this study cannot be put into the context of other

studies. Use of the Beck Depression Inventory is standard for assessing depression as a

current state. However, the study suffers from lack of information about what lifetime

psychiatric diagnoses the participants have. One would expect the access to empathetic

treatment providers would differ greatly among those with psychotic depression and those

without as just one example.

Response: In response to the reviewer’s comment that the instruments used do not test thestudy hypotheses, we would like to clarify that there are no study hypotheses for this paper.

We fear the reviewer is confusing the purpose of this submission with the parent randomised

controlled trial which has already been reported on, and from which participants in the current

study were recruited. The quantitative data is presented as context for the participants who

provided the individual interviews, but has indeed been collected using standardised,

validated measures as outlined in the methods.

Additionally, we note that this manuscript was submitted previous to the current submission

and was reviewed by two expert reviewers. At that point in time the decision was a ‘major

revision and resubmit’. We took some time to revise the submission, and thus submitted the

current manuscript as a new submission. However, we addressed all reviewer comments from

the prior submission, and uploaded a document indicating how this occurred with our

submission. This is also now provided below.

The paradox of engagement: The support needs of people with comorbid depression and

alcohol misuse who had previously attempted suicide

REVIEWER 1 – please see Reviewer Comments in bold and our response in normal text.

The authors present a well-written qualitative study of suicidal behavior among people

experiencing depression and alcohol/substance use. The analytic insights contain a few

quite promising leads, such as the “paradox of engagement,” wherein participants

wanted to be self-reliant yet expected others/health systems to reach out and help them.

The authors’ identification of complexities with regard to social support systems and

suggested use of alternative forms of support for individuals avoiding formal ones are

well-received. I do think, however, that the paper needs a bit more revision before being

suitable for publication.

We thank the Reviewer for this thoughtful reflection on the manuscript and for the comments

that follow. We trust we have addressed them adequately.

First, there is an issue with the nature and organization of findings. With qualitative

research of this kind that, as you mentioned, attempts to describe participants’

experienced meanings, the overarching themes should reflect such experienced

meanings as directly as possible. As it stands now, however, some of the themes do not

reflect their experienced meanings, but more seem to reflect the researcher’s frame and

understandable intention to enhance help-seeking behaviors. For instance, “mode of

help-seeking”, “barriers to help-seeking”, and “comorbidities” are themes that do not

appear to reflect participants’ directly experienced meanings. From my reading, it

appears that almost no one in this sample reached out for support, which would suggest

that there was little to no help-seeking involved (at least based on the data presented).

The realm of others were experienced more along the lines of shaming and dismissive,

with the participant feeling like a burden to them. These experiential parts were placed

under your heading of “barriers to help-seeking”, but it does not seem that, for many,

they tried to reach out and then experienced barriers (some did experience it this way,

but not all). For most, it seemed that there was a basic deprivation of support,

exacerbated by logistical, financial, and geographical structures, all of which presented a

context in which participants rarely considered help. Even the notion of “help” needs to

be explored more, perhaps with reference to the circumstances that led them to consider

suicide in the first place. I imagine that participants wanted “help” for many things,

including with aspects of their life circumstances (e.g., more help with childcare).

This is an important insight, and we thank the Reviewer for these reflections. We have gone

back to the data provided by participants and sought to more accurately apply thematic

meaning to their experiences as per their utterances, rather than from the higher order labels

that we (as healthcare providers and researchers) might use to categorise these experiences.

For example, we have reframed “mode of help-seeking” to “affordability, access, and

awareness of services”, “barriers to help-seeking” to “reluctance to seek help”, and

“comorbidities” to “compounding life events and coping mechanisms”. We have retained the

theme of “paradox of engagement in help seeking” and attempted to expand on this paradox.

We have attempted to represent, from the participants’ experience, what these themes

represent, rather than interpret (using our own frameworks) what we thought they were

referring to.

One main purpose of making these qualitative distinctions and of remaining close to the

original experience is to go beyond what is already conceptually known about the topic,

in order to uncover insights that may be currently obscured by our own frames of

theorizing. After all, it is clear that the mental health system was not a suitable option

for these participants, so research that remains closer to their actual experience is better

positioned to close the gap between everyday community life and mental health systems.

The section on paradox of engagement gets much, much closer to the way they

experience it and was a quite good section overall (the summary at the lead of the

Discussion was also quite good and near-to-experience in this respect). In all, though,

there needs to be much more attempt at reflecting the structure of the experience as

lived, replete with descriptions of essential meanings.

Again, we thank the reviewer for these valuable insights, and have tried to reframe the data

using the participants’ own utterances rather than our interpretation of what this relates to in

the broader healthcare literature and context. We have applied Bronfenbrenner’s ecological

systems theory to the interpretation of the main findings of the study, and have attempted to

ensure that this interpretative discussion is separated out from the results of the study.

Second, the concrete rationale for this specific study needs to be better developed. As of

now, it appears as a more global connection between previous literature and this

study—which is a bit too general for a journal of this type. One way to correct this is, I

believe, if the authors conduct a more concrete critique of the previous literature, which

would then set up the specific need for this current study. There are some gaps

otherwise: For instance, why focus on suicide attempts? Why focus on help received and

desired? It may seem obvious but filling in these gaps would greatly aid both the

organization and readability of your paper. On a related note, I find the following

statement hard to believe: “Despite this, little research exists on the relationship between

alcohol use and suicidal behaviours.” Most practicing clinicians are well-aware of this

link, as it is also reflected in most suicide prevention guidelines. Presumably these

guidelines come not only from years of clinical experience but also from research

(including case studies). I imagine many readers will also wonder about this, so I would

suggest that the authors at least mention some of the available research or guidelines,

and then suggest a reason as to why more research is needed.

We have added literature and a consideration of the nature of suicidality into the introduction,

and removed the statement about alcohol use and suicidal behaviours. To clarify, however,

whilst clear data have demonstrated the links between suicidal behaviours and alcohol misuse,

much less has examined help-seeking for suicidal ideation/attempts in people with alcohol

misuse problems. We have also presented a discussion of why more research is needed in this

area.

Once the above two issues are addressed, I believe that the authors will be in a place to

offer even more specified and novel suggestions for policy, practice, and action. Some of

the ones mentioned are of course good and thoughtful, such as the focus on alternative

means of support and ways to address the “burden” issue. But I believe more can be

said about ways to combat stigma, isolation, and hopelessness, such as access to stories

of recovery or the involvement of peers who have been through it. And again, we need to

know more about the life circumstances that are leading them to consider suicide in the

first place, which may ultimately also implicate the depriving social world around them.

Our interviews did not explore the life circumstances leading participants to consider suicide

in the first place, which is a limitation of the current study (reflected in the Discussion). We

have applied Ecosystems theory to the study results, which attempts to understand (and place)

the role of peers, family, and other support systems in the life of our participants leading up to

their suicide attempt. We believe that applying this lens deepens the insights gained from this

piece of work.

I would explain more about your view of what “data saturation” entails. I generally

remain a bit uncomfortable with this idea of saturation, given that more can always be

said or explored about a given topic, even within one participant. Further, your sample

appears to be quite acculturated to mainstream culture. So, I would personally avoid

suggesting that saturation means that your findings are all that can be said about the

topic.

We have removed the term ‘data saturation’ from the manuscript, as we agree with the

Reviewer’s comments in relation to the current sample. We have addressed the sample size as

a limitation in the revised manuscript.

On a related note, there’s a need to speak on limitations with regard to excluding people

with an ‘inability to comprehend English’. I’d imagine these folks are doubly suffering

from social isolation and are in great need of support.

This has indeed been added to the limitations section in the discussion.

Kindly clarify if the 4 standard drinks is the recommended limit or exceeds the limit—

the writing is a bit unclear.

Four standard drinks is the recommended limit and drining in excess of this limit was the

basis for inclusion in the study. This has been clarified.

There are a handful of typos which can be found upon another read-through.

We hope we have addressed these.

I was unsure what the restrictions on data access were, or if these need to be stated.

Thank you for this point. Given the limits of our ethics approval for the study, and the

identifiable nature of the interview transcripts, we are unable to provide open access to the

data for the current study. Instead, we can provide access to the transcripts by application via

our Human Research Ethics committee and have provided details about this accordingly.

Overall, this paper holds promise and is in a much-needed area of study. My suggestion

is for the authors to return to their participants’ experience again, to reflect its internal

structures more closely, all in the attempt to close the gap between those in need and the

supports that could perhaps help them.

Thank you for this comment. We hope that in doing this, we have been able to address (and

adequately reflect) the Reviewer’s comments on the manuscript.

---

## [Decision Letter · Decision Letter 1]

23 Jan 2020

PONE-D-19-16846R1

The paradox of engagement: The support needs of people with comorbid depression and alcohol misuse who had previously attempted suicide

PLOS ONE

Dear Professor Kay-Lambkin,

Thank you for submitting your manuscript to PLOS ONE. After careful consideration, we feel that it has merit but does not fully meet PLOS ONE’s publication criteria as it currently stands. Therefore, we invite you to submit a revised version of the manuscript that addresses the points raised during the review process.

Please, be advised that submitting a revision does not guarantee acceptance.

We would appreciate receiving your revised manuscript by Mar 08 2020 11:59PM. To enhance the reproducibility of your results, we recommend that if applicable you deposit your laboratory protocols in protocols.io, where a protocol can be assigned its own identifier (DOI) such that it can be cited independently in the future. For instructions see: http://journals.plos.org/plosone/s/submission-guidelines#loc-laboratory-protocols

We look forward to receiving your revised manuscript.

Kind regards,

Vincenzo De Luca

Catherine Haighton, PhD

Academic Editors

PLOS ONE

Journal Requirements:

2) Please include captions for your Supporting Information files at the end of your manuscript, and update any in-text citations to match accordingly. Please see our Supporting Information guidelines for more information: http://journals.plos.org/plosone/s/supporting-information.

Reviewers' comments:

Reviewer's Responses to Questions

**Comments to the Author**

1. If the authors have adequately addressed your comments raised in a previous round of review and you feel that this manuscript is now acceptable for publication, you may indicate that here to bypass the “Comments to the Author” section, enter your conflict of interest statement in the “Confidential to Editor” section, and submit your "Accept" recommendation.

Reviewer #2: (No Response)

2. Is the manuscript technically sound, and do the data support the conclusions?

Reviewer #2: Partly

3. Has the statistical analysis been performed appropriately and rigorously? 

Reviewer #2: I Don't Know

4. Have the authors made all data underlying the findings in their manuscript fully available?

Reviewer #2: Yes

5. Is the manuscript presented in an intelligible fashion and written in standard English?

Reviewer #2: Yes

6. Review Comments to the Author

Reviewer #2: Challenges and ambivalence in help-seeking for suicidal individuals with comorbid depression and alcohol use disorder is a clinically relevant and complex topic. I applaud the authors of this paper for trying to delve deeper into this topic using qualitative based research methodology. In my opinion, the main flaw of this paper is the attempt of the authors to make general inferences from very limited qualitative data to support their theory in the Discussion and Conclusion sections. This paper would be stronger if the authors stuck more closely to their qualitative data without trying to make larger inferences. Below area a few examples that illustrate my points:

1. I think it is appropriate to use a theoretical lens as you do (Ie. ecological systems theory) to interpret your findings, but some caution is warranted when you try to use your qualitative data to justify your theoretical lens in both the Discussion and Conclusion. For example, at the end of the Conclusion you state that “Application of an ecological systems perspective confirmed the significance of support at the microsystemic level. It also illuminated the critical link between attitudes and behaviours at the micro level, and constructions of suicide at the broader macro level. Recognising and addressing the cascading influence of broader socio-cultural perspectives on suicide, mental health and comorbidity, is crucial to ensuring the effectiveness of future intervention and prevention measures. ” I do not think your qualitative data “confirms” these conclusions and I did not see any evidence from the data to support a “critical link” between the micro and macro level. Perhaps you could say something like: “the qualitative data revealed themes that illustrated many challenges of engaging with different support systems which lends itself well to an ecological systems theory perspective when considering interventions and approaches for this complex population.”

2. Similarly, in the last paragraph of the Discussion Section, you make statements that are too broad and do not justify the data, e.g.: “The key finding of this study, that individuals experiencing acute phases of suicidality do not typically access traditional treatment services and are reluctant to seek support from family and friends, is consistent with findings from a number of previous studies (Fogarty et al. 2018). While this finding highlights the importance of support and engagement at the micro and meso-systemic levels, it also illuminates the critical role of broader social and cultural macrosystemic – factors in shaping people’s attitudes and behaviours” I do not think that is the “key finding” of this study based on the qualitative data you presented. In my opinion, your study illustrates the ambivalence and different types of challenges people experience seeking help when they are suicidal when they engage with different systems of support. In addition, you state in an earlier paragraph in the Discussion Section that people are more likely to seek help from family and friends rather than professionals which contradicts this last paragraph where you introduce the “key finding” that e people are reluctant to seek help from family and friends.

3. I don’t find the concept “paradox of service engagement” very helpful and I don’t think it does justice to your data. My impression from your data is that many of these individuals had significant ambivalence asking for help due several factors such as negative experiences with family or formal providers or fears about how these different groups would respond if they reached out for help and disclosed being suicidal. I would prefer more descriptive words in your title and abstract that are closer to the data such as the words “ambivalence” or “challenges” rather than the impression that you have discovered a whole new concept called “paradox of service engagement” – again similar to the Discussion and Conclusion sections it seems like you are trying to reify concepts from qualitative data which does not lend itself to such generalizations.

7. PLOS authors have the option to publish the peer review history of their article (what does this mean?). If published, this will include your full peer review and any attached files.

Reviewer #2: No

---

## [Author Response · Author response to Decision Letter 1]

26 Feb 2020

Response to Reviewers:

Thank you for taking the time to review our manuscript and provide such thoughtful feedback on our work, which has helped to strengthen our paper. Please see our response to each point raised by the reviewer below. We welcome any further feedback on our revisions.

Reviewer #2: In my opinion, the main flaw of this paper is the attempt of the authors to make general inferences from very limited qualitative data to support their theory in the Discussion and Conclusion sections. For example,

At the end of the Conclusion you state that “Application of an ecological systems perspective confirmed the significance of support at the microsystemic level. It also illuminated the critical link between attitudes and behaviours at the micro level, and constructions of suicide at the broader macro level. Recognising and addressing the cascading influence of broader socio-cultural perspectives on suicide, mental health and comorbidity, is crucial to ensuring the effectiveness of future intervention and prevention measures.” I do not think your qualitative data “confirms” these conclusions and I did not see any evidence from the data to support a “critical link” between the micro and macro level. Perhaps you could say something like: “the qualitative data revealed themes that illustrated many challenges of engaging with different support systems which lends itself well to an ecological systems theory perspective when considering interventions and approaches for this complex population.”

Author response: Thank you. We agree with your feedback and have amended our wording in line with your suggestion. 

Reviewer #2: Similarly, in the last paragraph of the Discussion Section, you make statements that are too broad and do not justify the data, e.g.: “The key finding of this study, that individuals experiencing acute phases of suicidality do not typically access traditional treatment services and are reluctant to seek support from family and friends, is consistent with findings from a number of previous studies (Fogarty et al. 2018). While this finding highlights the importance of support and engagement at the micro and meso-systemic levels, it also illuminates the critical role of broader social and cultural macrosystemic – factors in shaping people’s attitudes and behaviours” I do not think that is the “key finding” of this study based on the qualitative data you presented. In my opinion, your study illustrates the ambivalence and different types of challenges people experience seeking help when they are suicidal when they engage with different systems of support. In addition, you state in an earlier paragraph in the Discussion Section that people are more likely to seek help from family and friends rather than professionals which contradicts this last paragraph where you introduce the “key finding” that e people are reluctant to seek help from family and friends.

Author response: Thank you for making us aware of these issues. We have revised this paragraph to address the inaccuracies and the contradicting statements you refer to (see p. 25).

Reviewer #2: I don’t find the concept “paradox of service engagement” very helpful and I don’t think it does justice to your data. My impression from your data is that many of these individuals had significant ambivalence asking for help due several factors such as negative experiences with family or formal providers or fears about how these different groups would respond if they reached out for help and disclosed being suicidal. I would prefer more descriptive words in your title and abstract that are closer to the data such as the words “ambivalence” or “challenges” rather than the impression that you have discovered a whole new concept called “paradox of service engagement” – again similar to the Discussion and Conclusion sections it seems like you are trying to reify concepts from qualitative data which does not lend itself to such generalizations.

Author response: Upon reflection, we agree that the “paradox of engagement” does not do justice to our data. We have removed this expression from the manuscript (including from the title and abstract) and have instead used descriptive words that are closer to the data (see p. 1, 2, 19, 23, 26).

---

## [Decision Letter · Decision Letter 2]

30 Mar 2020

Understanding ambivalence in help-seeking for suicidal people with comorbid depression and alcohol misuse

PONE-D-19-16846R2

Dear Dr. Kay-Lambkin,

We are pleased to inform you that your manuscript has been judged scientifically suitable for publication and will be formally accepted for publication once it complies with all outstanding technical requirements.

With kind regards,

Vincenzo De Luca

Academic Editor

PLOS ONE

Additional Editor Comments (optional):

Reviewers' comments:

Reviewer's Responses to Questions

**Comments to the Author**

1. If the authors have adequately addressed your comments raised in a previous round of review and you feel that this manuscript is now acceptable for publication, you may indicate that here to bypass the “Comments to the Author” section, enter your conflict of interest statement in the “Confidential to Editor” section, and submit your "Accept" recommendation.

Reviewer #2: All comments have been addressed

2. Is the manuscript technically sound, and do the data support the conclusions?

Reviewer #2: Yes

3. Has the statistical analysis been performed appropriately and rigorously? 

Reviewer #2: N/A

4. Have the authors made all data underlying the findings in their manuscript fully available?

Reviewer #2: Yes

5. Is the manuscript presented in an intelligible fashion and written in standard English?

Reviewer #2: Yes

6. Review Comments to the Author

Reviewer #2: (No Response)

7. PLOS authors have the option to publish the peer review history of their article (what does this mean?). If published, this will include your full peer review and any attached files.

Reviewer #2: Yes: Jan Malat

---

## [Editor Report · Acceptance letter]

2 Apr 2020

PONE-D-19-16846R2 

Understanding ambivalence in help-seeking for suicidal people with comorbid depression and alcohol misuse 

Dear Dr. Kay-Lambkin:

I am pleased to inform you that your manuscript has been deemed suitable for publication in PLOS ONE. Congratulations! Your manuscript is now with our production department. 

With kind regards,

on behalf of

Dr. Vincenzo De Luca 

Academic Editor

PLOS ONE